# Assisted sexual coral recruits show high thermal tolerance to the 2023 Caribbean mass bleaching event

Margaret W. Miller[1]*, Sandra Mendoza Quiroz[1,2], Liam Lachs[3], Anastazia T. Banaszak[2], Valérie F. Chamberland[1,4,5], James R. Guest[3], Alexandra N. Gutting[6], Kelly R. W. Latijnhouwers[1,4,5], Rita I. Sellares-Blasco[7], Francesca Virdis[8], Maria F. Villalpando[7], Dirk Petersen[1]

1 SECORE International, Miami, FL, United States of America, 2 Unidad Académica de Sistemas Arrecifales, Universidad Nacional Autónoma de México, Puerto Morelos, Quintana Roo, México, 3 School of Natural and Environmental Sciences, Newcastle University, Newcastle upon Tyne, United Kingdom, 4 CARMABI Foundation, Willemstad, Curaçao, 5 Institute for Biodiversity and Ecosystem Dynamics, University of Amsterdam, Amsterdam, The Netherlands, 6 The Nature Conservancy, St. Croix, US Virgin Islands, 7 Fundación Dominicana de Estudios Marinos (FUNDEMAR), Bayahíbe, Dominican Republic, 8 Reef Renewal Foundation Bonaire (RRFB), Bonaire, Caribbean Netherlands

* m.miller@secore.org

**Data Availability Statement:** All relevant data are within the manuscript and its Supporting Information files.

## Abstract

Assisted sexual coral propagation, resulting in greater genet diversity via genetic recombination, has been hypothesized to lead to more adaptable and, hence, resilient restored populations compared to more common clonal techniques. Coral restoration efforts have resulted in substantial populations of 'Assisted sexual Recruits' (i.e., juvenile corals derived from assisted sexual reproduction; AR) of multiple species outplanted to reefs or held in *in situ* nurseries across many locations in the Caribbean. These AR populations provided context to evaluate their relative resilience compared to co-occurring coral populations during the 2023 marine heat wave of unprecedented duration and intensity that affected the entire Caribbean. Populations of six species of AR, most ranging in age from 1–4 years, were surveyed across five regions during the mass bleaching season in 2023 (Aug-Dec), alongside co-occurring groups of corals to compare prevalence of bleaching and related mortality. Comparison groups included conspecific adult colonies as available, but also the extant co-occurring coral assemblages in which conspecifics were rare or lacking, as well as small, propagated coral fragments. Assisted sexual recruits had significantly lower prevalence of bleaching impacts (overall pooled ~ 10%) than conspecific coral populations typically comprised of larger colonies (~ 60–100% depending on species). In addition, small corals derived from fragmentation (rather than sexual propagation) in two regions showed bleaching susceptibility intermediate between AR and wild adults. Overall, AR exhibited high bleaching resistance under heat stress exposure up to and exceeding Degree Heating Weeks of 20°C-weeks. As coral reefs throughout the globe are subject to increasingly frequent and intense marine heatwaves, restoration activities that include sexual reproduction and seeding can make an important contribution to sustain coral populations.

**Funding:** Funding support was received from The Builders Initiative (to DP; https://www.buildersinitiative.org/), the Caribbean Biodiversity Fund (to DP and RSB via the Ocean Foundation; https://caribbeanbiodiversityfund.org/), The National Fish and Wildlife Foundation (Award ID0318.20.069532 to J.Ward, The Nature Conservancy; nfwf.org), BMUV IKI Coral Carib 2023 (to RSB; https://www.international-climate-initiative.com/en/), Consejo Nacional de Humanidades, Ciencias y Tecnologías (Project #425888 to ATB; https://conahcyt.mx/), State government of Quintana Roo, Mexico, (Project # 2021 Z4 to ATB) These funders played no role in the study design, data collection and analysis, decision to publish, or preparation of the manuscript.

**Competing interests:** The authors have declared that no competing interests exist.

# Introduction

The rapid pace of coral population and species declines combined with the inadequacy of actions to address their root causes (e.g., greenhouse gas emissions and local pollution) is driving increased interest and investment in active coral restoration globally. While coral restoration alone is not a solution to the current coral crisis, it may 'buy time' for crucial efforts to address environmental degradation [1]. Asexual propagation (i.e., through fragmentation) has been widely utilized in fast-growing, branching species for nearly two decades as a means of rapidly producing corals for transplantation [2, 3]. In contrast, sexual propagation (or breeding) has only more recently been widely pursued due to additional challenges involving timing, multiple steps required and resultant need for additional expertise and facilities in this process [4, 5]. High natural fecundity of spawning corals provides great potential for upscaling while genetic recombination during sexual reproduction provides the expectation of higher adaptive potential to rapidly changing environments in resulting restored populations [6]. In the Caribbean, the major reef-building species (*Acropora* spp., *Orbicella* spp., and to a lesser extent, brain corals) are broadcast spawners whose natural sexual recruitment has remained poor to non-existent [7–10] and hence have long been key targets for breeding and assisted recruitment efforts [11–13].

Corals generally experience very high mortality rates in the period immediately after settlement, being particularly vulnerable to competition and predation [e.g., 14, 15], and highly susceptible to disease [16]. Meanwhile, previous work has suggested a general pattern that juvenile corals (often estimated simply as small colonies in wild populations) are less susceptible to heat-related bleaching. For example, Brandt [17] found large colonies of *Colpophyllia natans* were more susceptible to thermal bleaching, succumbing at an accumulated heat stress level of 3–4 Degree Heating Weeks (DHW, units˚C-weeks; [18]) whereas small colonies did not bleach until 9–10˚C-weeks. It has been hypothesized that juvenile corals may be more resistant to heat stress due to a number of mechanisms such as living in shaded environments, different mass transfer rates, or more flexible symbiont associations.

Throughout the Caribbean region, 2023 was a record-breaking year in terms of heat stress with accumulation in coral reef habitats of not just the highest on record (Fig 1), but in many

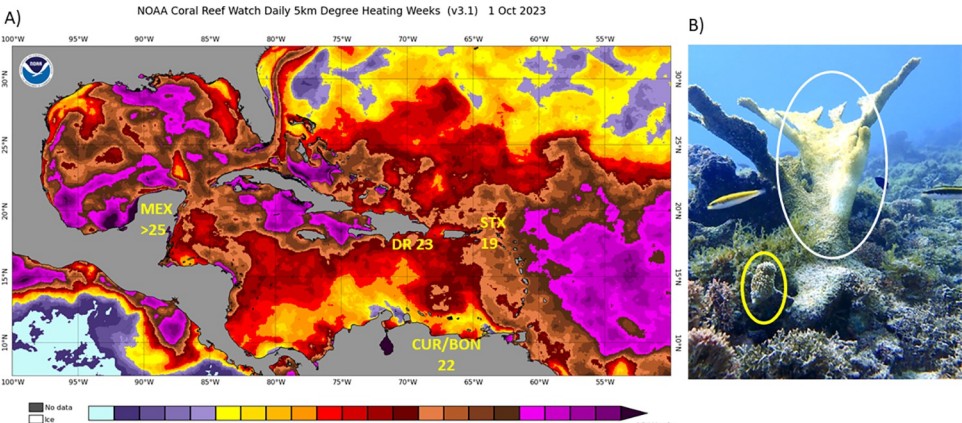

**Fig 1.** A) Satellite-derived estimates of heat stress exposure (Degree Heating Weeks, as of 1 Oct 2023; from https://coralreefwatch.noaa.gov/data/5km/v3.1_op/image/daily/dhw/png/2023/10/ct5km_dhw_v3.1_caribbean_20231001.png) across the Caribbean with the estimated maxima for the five study regions (MEX: Puerto Morelos, Mexico; DR: Bayahíbe, Dominican Republic; STX: St. Croix, US Virgin Islands; CUR: Curaçao; BON: Bonaire). B) Image of bleaching-resistant Assisted Recruit (red circle) adjacent to bleaching or recently dead adult conspecific (white circle) of *Acropora palmata* (Dominican Republic; photo M.Villalpando).

cases *four times* the previous highest record and lasting for one to two months longer than normal seasonal patterns [19]. The relatively long duration of efforts toward coral restoration via breeding and assisted recruitment in this region [11–13] thus provided an opportunity to evaluate the resistance of sexually propagated juveniles under extreme levels of thermal stress. We sought to quantify the bleaching response of Assisted Recruits (AR; derived from coral restoration activities involving sexual propagation and seeding) of six reef-building coral species across five regions spanning the Caribbean basin, alongside that of various comparison groups of corals, including co-occurring 'wild' assemblages, conspecific corals at parental sites (from which gametes were collected), and small colonies propagated via fragmentation.

## Materials & methods

Surveys were conducted of cohorts of assisted coral recruits created in five regions between 2011 and 2022 (most between 2019–2022), all of which were derived from coral restoration programs implementing sexual propagation and seeding (Table 1). In some regions, these cohorts consisted of individually tagged corals that were surveyed repeatedly as part of performance monitoring. In other cases, surveys were conducted in areas where unmarked AR had been outplanted haphazardly but were recognizable because they were attached to artificial settlement substrates. All were in areas where natural recruitment of these spawning species is rare or absent. Each colony was visually scored as having one of four conditions: 'healthy'

**Table 1. Information for the surveyed Assisted Recruits (AR; derived from coral breeding efforts for restoration) and various comparison populations (WC: Wild conspecific adults at the same site as AR; PAR: Parental site conspecifics; WA: Wild adult assemblage at the same site as AR; WJ: Wild juvenile assemblage at same site as AR; F: small colonies derived from fragment (asexual) propagation) across the five regions (MEX: Puerto Morelos, Mexico; DR: Bayahibe, Dominican Republic; STX: St. Croix, US Virgin Islands; CUR: Curacao; BON: Bonaire).** Maximum Degree Heating Weeks (DHW, units˚C-weeks) estimated from nearest NOAA Coral Reef Watch 'virtual stations', https://coralreefwatch.noaa.gov/product/vs/data.php). Ofav: *Orbicella faveolata*; Oann: *O. annularis*; Dlab: *Diploria labyrinthiformis*; Apal: *Acropora palmata*; Cnat: *Colpophyllia natans*; Pstr: *Pseudodiploria strigosa*.

| Region/Site | Focal Species (AR/PAR) | Population | Depth (m) | Lat | Lon | Survey Date (2023) | Max DHW (Timing) |
|---|---|---|---|---|---|---|---|
| **MEX/Jardines** | Ofav, Oann, Dlab, Pstr | AR, WA, WJ PAR(Dlab) | 6 | 20.8314 | -86.8745 | 23–31 Aug 26 Sep (WJ) | >25 (Sept/Oct) |
| **MEX/Acuario** | Dlab | AR, WA, WJ | 12 | 20.8059 | -86.8790 | 19 Sept | |
| **MEX/Bonanza** | Dlab | PAR | 9 | 20.9641 | -86.8073 | 11 Aug | |
| **MEX/Limones** | Apal | PAR (2022)* | 2.5 | 20.9885 | -86.7972 | 30 Aug | |
| **MEX/Bocana** | Apal | PAR (2011–15)* | 6 | 20.8740 | -86.8513 | 25–30 Aug | |
| **MEX/Picudas** | Apal | AR (2011–15)*, WC | 3 | 20.8833 | -86.8483 | 30 Aug | |
| **MEX/Punta Maroma** | Apal | AR (2022)*, WA,WJ | 3–5 | 20.7156 | -86.9741 | 11–12 Sept | |
| **STX/Channel Rock** | Dlab, Pstr | AR AR, WC | 8 | 17.7667 | -64.5946 | 30 Nov 7 Nov | 19 (Late Oct) |
| **STX/Cane Bay East** | Dlab | PAR | 7.3 | 17.7742 | -64.8116 | 31 Oct | |
| **STX/Deep End Beach** | Pstr | PAR | 4 | 17.7614 | -64.6693 | 2 Nov | |
| **STX/Cane Bay West** | Pstr | WC | 10 | 17.7736 | -64.8137 | 31 Oct | |
| **CUR/ CARMABI** | Cnat | AR, WC | 5–10 | 12.1213 | -68.9697 | 20 Nov | 22 (Nov) |
| **BON/Buddy's** | Cnat Ofav/Oann | AR F | 5 | 12.1712 | -68.2891 | 28 Nov | 22 (Nov) |
| **DR/CRIB site** | Dlab | AR,WA,WJ | 3–4 | 18.3715 | -68.8478 | 4 Dec | 23 (Late Oct/ early Nov) |
| **DR/Sombrero** | Apal | AR,WA,WJ, F, | 1–2 | 18.3706 | -68.8467 | 4 Dec | |

*Gametes for AR of *Acropora palmata* in MEX were collected at separate parental sites in different time frames

coloration, 'pale', fully 'bleached' (appearing completely white over at least a portion of the live colony surface), or recently dead (whole-colony mortality with stark white skeleton, and/or recorded mortality of tagged individuals within the previous four months). Timing of surveys was different in each location, but was within four weeks of estimated peak heat stress for that location based on NOAA's Coral Reef Watch satellite-derived heat stress estimates (https://coralreefwatch.noaa.gov/product/vs/data.php) as described in Table 1.

In each location, one or more 'comparison' populations were also surveyed, scoring each colony with the same aforementioned health status categories. Because of the rarity of conspecifics in many current Caribbean reef assemblages, we sampled a variety of different comparison populations of co-occurring corals among the different regions to maximize inclusion of conspecifics. In a few cases, 'wild' conspecific adults (WC) were present in the same reef site as the outplanted AR and were targeted for scoring via directed search. This was considered the most appropriate comparison when both species and macro-environmental exposure were matched at the same site. In other regions (Mexico and Dominican Republic) we conducted standardized surveys (AGRRA method as described in [20] or [21]) of the extant 'wild' adult (WA; > 4 cm diameter) and 'wild' juvenile (WJ; < 4cm) assemblages in the reef surrounding the outplanted AR. Four cm diameter was used as a demographic cutoff, consistent with widespread monitoring protocols (e.g., AGRRA, Lang et al. 2012; NOAA's National Coral Reef Monitoring Program, https://ncrmp.coralreef.noaa.gov/pages/ncrmp-data#BenthicSection), but we acknowledge that different species (and different individual colonies) may mature at somewhat smaller or larger sizes. These surveys were intended to provide comparisons within the same macro-environmental exposure, but these assemblages were generally mis-matched to the AR species. In several cases, surveys of conspecific colonies at parental sites (PAR) were also undertaken; i.e., gamete collection sites from which the AR were bred, providing within-species comparison but at different sites (hence, different macro-environmental exposures). Lastly, in two regions, small colonies propagated by fragmentation (F) that were co-located with AR either in an *in situ* nursery (Bonaire, heterospecific comparison colonies) or within the same outplant site (Dominican Republic, conspecifics) were also surveyed.

All pooled comparison populations were compared to AR populations qualitatively (i.e. visually via stacked bar graphs). However, we restricted our statistical analyses to conspecific comparisons within the six species used for restoration (AR species listed in Table 1) to remove potential confounding effects of differential species susceptibility. That is, data were subsetted from all comparison population types to include only individuals of these six species (i.e., 303 observations of 1176 total Comparison observations). Statistical analyses were run to assess whether the severity of bleaching responses differed between AR versus conspecific colonies from Comparison populations (fixed factor with two levels, herein referred to as AR-Comp). We tested the effect of this fixed factor (Assisted Recruit vs. Comparison; AR-Comp) on the ordinal health status scores (0: healthy; 1: pale; 2: bleached; 3: recently dead) by species using a cumulative link model (CLM; ordinal regression, R-package: ordinal) in the form: Status ~ AR-Comp * Species, with species-specific effects computed using least-squares means.

Given the complexity of possible confounding factors in this dataset (multiple regions, sites, types of comparison population, and survey methods, with each factor combination being represented unequally by species), including them as random effects in the CLM was not compatible with the modelling framework. Therefore, we conducted two further statistical models to test if the CLM results were robust to possible confounding factors. First, we converted the ordinal health status response into a Bleaching and Mortality Index (BMI), with scores set to equidistant values between 0 (healthy) and 1 (recently dead), and ran a Generalized Linear Model (GLM) with a binomial error distribution equivalent to the CLM (form: BMI ~ AR-Comp * Species). The assumption of numerical equidistance between health status

categories was tested by comparing the CLM and GLM results. Second, the species-specific effect of AR-Comp on BMI was tested using a Generalized Linear Mixed-effect Model (GLMM; R-package: glmmTMB) with binomial error distribution, accounting for differences among regions (5-level random effect), sites (14-level random effect), and comparison population types (6-level random effect) by fitting random intercepts for each (BMI ~ AR-Comp * Species + (1|Region) + (1|Site) + (1|Population.type)). Notably, the species that showed statistically significant difference in BMI between AR and other colonies (i.e. significant AR-Comp effect) in the CLM were also statistically significant in the GLM and GLMM, although with larger uncertainties (and hence higher $P$ values, S1 Table). Therefore, all reported $P$ values in the remainder of the manuscript refer to the GLMM. Lastly, we visually compared the frequency of bleaching states in the AR populations to subsets of the various comparison populations described above to explore specific hypotheses for increased thermal tolerance of AR.

## Results and discussion

The pooled sample of all AR was over three times more likely to appear visually healthy (i.e., normal coloration) during the 2023 marine heatwave, with a 90% prevalence of healthy-colored AR (695/767), in comparison to only 24% (286/1176 colonies) in the pooled sample of all other colonies (Fig 2A). However, this summary pools unbalanced comparisons among species, sites (including likely differential macro-environmental exposure including thermal stress), and colony origin (e.g., fragment propagation vs. 'wild' colonies). Substantial differences are also apparent within species (Fig 2B), and confirmed by a generalized linear mixed effect model which does account for variability in bleaching due to possible confounding

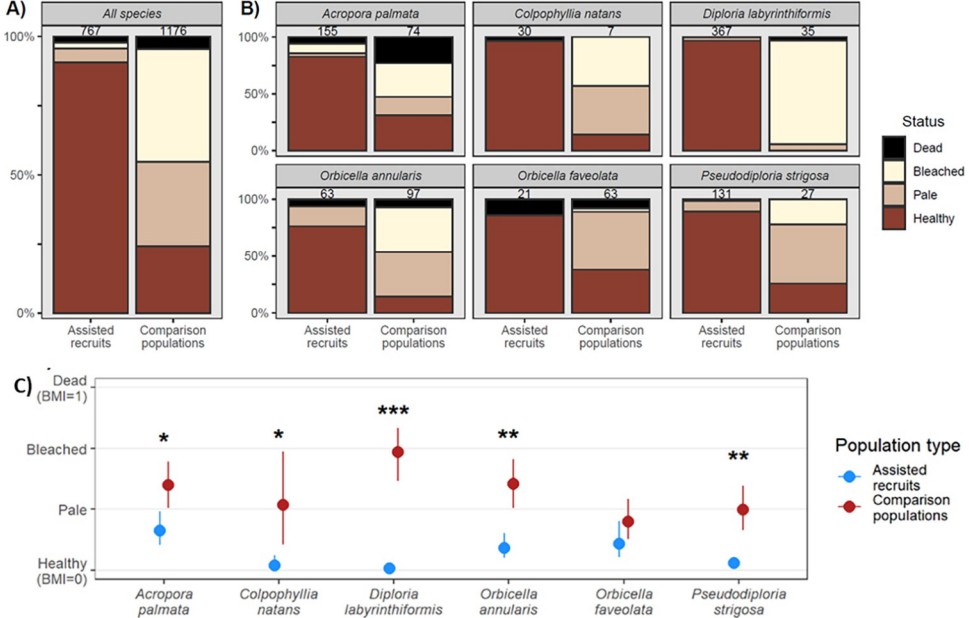

**Fig 2. Bleaching and mortality responses for the six species propagated for restoration.** A) Cumulative colony condition for Assisted Recruits (AR; derived from coral breeding efforts for restoration) and all comparison population types pooled across sites. Sample size for each group is given above the stacked bars. B) Subsets of data from A) that are parsed by the six species for which AR existed. C) Differences in Bleaching and Mortality Index (BMI) between AR and conspecific comparison colonies. Estimates of the mean BMI (points) and SE (bars) are based on a binomial GLMM in the form: BMI ~ AR-Comp * Species, accounting for differences among regions, sites, and comparison population type (random intercepts). Statistical significance of pairwise comparisons by species are shown for $P < 0.05$ (*), $P < 0.01$ (**), and $P < 0.001$ (***).

effects of region, site, and comparison population type. These GLMM analyses confirm that AR for each species except *O. faveolata* showed significantly greater resistance compared to conspecific Comparison corals (GLMM post-hoc comparisons, *P* values < 0.05, Fig 2C). *O. faveolata* AR do appear significantly more heat resistant in the CLM (ordinal) model (S1 Table). However, when accounting for additional confounding factors such as location, comparison population type, etc. in the GLMM, increased uncertainty due to limited observations (AR were from a single site in Mexico while available Comparison colonies were only Fragments from Bonaire) yields larger confidence intervals and hence lack of significance in the AR-Comp bleaching comparison (Fig 2C).

It is worth noting that most (65%) of the comparison colonies that were scored in systematic surveys of coral assemblages surrounding AR outplants (in the Dominican Republic (2 sites) and Mexico (3 sites)) were comprised of distinct species (*Porites* spp, *Agaricia* spp, and *Siderastrea siderea*). The pooled frequency of colonies with bleaching signs in these extant assemblages was 78% for adults (>4 cm; 170/767) and 60% for juveniles (<4 cm; 58/144); so similar susceptibility for adults and higher susceptibility for juveniles than generally observed in the six AR species. The relative dearth of these six hermaphroditic spawning, reef building species, and the virtual absence of their natural recruits are compelling reasons that restoration is needed for these species; the increased bleaching resistance of AR compared to naturally occurring juveniles of primarily weedy species makes sexual propagation an important method of restoration.

One caveat relates to the potential artifact of comparing (whole colony) mortality between large and small colonies, especially in the context of multiple stressors. Small colonies (including AR) might succumb to lethal conditions (such as bleaching) more quickly than large colonies. Therefore, bleaching impact as whole-colony mortality of AR colonies might be underestimated (relative to larger, slower-dying colonies) in one-off surveys because they may have died sooner and are no longer detectable due to overgrowth. This potential underestimate is at least partially offset as many of the AR scored in the study were individually tagged and therefore their mortality over the previous monitoring interval (up to 4 months) was not overlooked. Conversely, the bleaching mortality, particularly of small colonies might have been overestimated due to confounding mortality from disease. Stony coral tissue loss disease (SCTLD) was indeed at high prevalence during the time of our surveys on reefs in Curaçao and Bonaire though data reported from Bonaire were from a nursery setting where active disease was not concurrently observed. Active SCTLD was observed in surveyed colonies in Curaçao, with two *C. natans* (conspecific) adults showing signs of active disease (but no bleaching).

## Is bleaching resistance a general characteristic of small and/or young colonies?

Results presented here are consistent with previous reports that juvenile corals are generally more bleaching resistant to moderate heat exposures (i.e., DHW of ~4–10˚C-weeks) than adult corals. For example, Speare et al. [22] found that heatwave-induced mortality was substantially lower for smaller size classes from the 2019 marine heatwave in Moorea which reached DHWs of 5–6˚C-weeks. Similarly, Brandt [17] reported reduced bleaching susceptibility in small corals (<10 cm diameter) compared to larger conspecifics, but only up to a DHW of 9˚C-weeks, after which all size classes converged at >75% bleaching prevalence. During the 2020 mass bleaching event in eastern Australia, Burn et al. [23] also found that patterns of relative bleaching susceptibility were dependent on the overall intensity of reef-wide mass bleaching among sites (used as a proxy of a reef's DHW exposure). For example, in eight of ten taxa, juveniles were less likely to bleach than adults, but this pattern disappeared at sites with

extremely high bleaching prevalence (>81% of colonies bleached). Although these authors do not provide DHW exposures across their sampling sites, NOAA satellite data suggests the highest levels experienced in this region in 2020 were in the range of 13–15˚C-weeks (https://coralreefwatch.noaa.gov/data/5km/v3.1_op/image/daily/dhw/png/2020/03/ct5km_dhw_v3.1_gbr_20200315.png), substantially lower than that experienced in the Caribbean 2023 event reported here. Thus, our study adds to previous reports by 1) providing observations on known, sexual recruits (rather than presumed recruits based on small size) of multiple species from restoration activities and 2) documenting patterns in the midst of a regional thermal stress event of unprecedented severity and duration, with DHW exposures estimated between 19 and 25˚C-weeks and durations in excess of five months across all study sites (Table 1).

While the AR in our study showed strong resistance to bleaching (90% of 771 colonies with healthy coloration), extant wild juvenile assemblages sampled at outplant sites (largely comprised of different species from those used for restoration) showed variable responses; relatively high tolerance in the Dominican Republic (80% of 25 colonies healthy) and much less tolerance in Mexico (only 32% of 119 colonies healthy). Colony size data were not systematically collected in our study (only available for a relatively small subset of conspecific colonies), but those available showed no significant relationship between colony size and BMI (S1 Fig), with the exception of *A. palmata* comparison populations where the anomalous trend showed better health status for larger colony sizes (S1 Fig).

In the two regions where we surveyed nearby small coral colonies propagated for restoration via fragmentation, they fared worse than the AR, in line with the fact that these fragments, although small in size, are not true juveniles. In Bonaire, *Colpophyllia natans* AR showed no bleaching, whereas small propagated fragments of *Orbicella* spp. (2–4 cm in diameter) co-located in a field nursery showed substantial prevalence of heat stress (only 22% of 143 colonies with healthy coloration). However, AR of *Orbicella* also showed slightly higher bleaching susceptibility than AR of *C. natans* (Fig 2A and 2B). Meanwhile, in a conspecific comparison, slightly larger (5–10 cm) fragment-originated colonies of *A. palmata* outplanted to Sombrero reef in the Dominican Republic showed intermediate levels of tolerance (between AR and other adults with 55% of 11 colonies appearing healthy; Fig 3A).

It is possible that the high bleaching resistance of sexually propagated juvenile colonies recorded in our study is lost when they reach adult stages later in life. Our dataset includes one cohort of >10 year-old (i.e., reproductive adult) colonies derived from AR of *A. palmata* in Mexico [13]. In contrast to the overall sample of AR and to the conspecific juvenile AR in the same location, this set of colonies was highly affected by the heat stress with none retaining healthy coloration and 32% (of 19 colonies) having already succumbed to bleaching-related mortality at the time of survey (Fig 3B). Adult colonies from their parental population also showed high bleaching related mortality (Fig 3B, 74% of 19 colonies). In comparison, a cohort of <2 year-old *A. palmata* AR outplanted in Mexico and their parent population showed the typical juvenile resistance (96% of 97 colonies healthy) and adult sensitivity (44% of 27 colonies healthy, Fig 3C). Together these observations suggest that thermal tolerance of coral AR is an ontogenic characteristic that diminishes with age.

Overall, some degree of bleaching resistance appears likely in both assisted and wild juvenile corals, but less so for small corals propagated through fragmentation (as they are derived from adult colonies). Nonetheless, on Caribbean reefs where juveniles of most reef-building coral species are virtually absent (and likely to be increasingly rare in other reef areas around the globe as global warming proceeds, e.g., [24]), interventions to assist coral recruitment can be an important tool to buffer population impacts (via persistence of young colonies) from heatwave-induced bleaching.

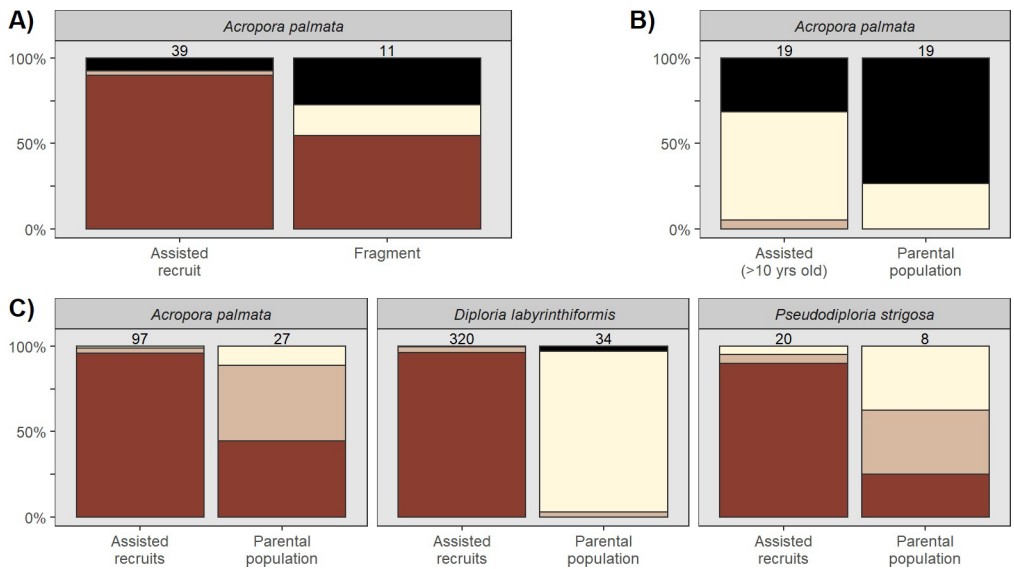

**Fig 3. Summary of bleaching and mortality responses of subsets of data relating to specific hypotheses.** A) *A. palmata* Assisted Recruits (AR) compared to small *A. palmata* colonies propagated by fragmentation at the same site in Dominican Republic. B) One cohort of adult *A. palmata* derived from AR in Mexico compared to its parental site. C) Comparing juvenile AR with colonies from their parental sites (PAR) for three species where the comparison was possible (*Acropora palmata* from Mexico, *Diploria labyrinthiformis* pooled from Mexico and St. Croix, and *Pseudodiploria strigosa* from St. Croix) is not consistent with the hypothesis that thermal tolerance is inherited. Sample size for each group is given above the stacked bar.

## Why are recruits more bleaching resistant?

This study was not designed to resolve the different mechanism(s) driving the emergent pattern of widespread bleaching resistance in AR. Several hypotheses are posed in the literature that may (partially) account for the observed pattern. Mumby [25] had proposed that wild recruits suffered less from a mass bleaching event due to their natural preference for settling in cryptic habitats, protecting them from exacerbating light stress. It is indeed possible that the AR in the current study experienced less light than comparison colonies. Although they were on artificial substrates, these substrates were outplanted by wedging them into reef crevices, and AR survive preferentially in grooves or other cryptic features of designed substrates [26, 27]. Another theory is that because juveniles are not engaging in gametogenesis that they enjoy greater physiological buffering from energy reserves than reproductive adults that are devoting energy to egg production [28]. Meanwhile, generally small colony size may simply help alleviate physiological stress by providing for greater mass-transfer across the colony surface [29]. These latter two hypotheses are consistent with the subsets of our dataset showing higher susceptibility of the small sample of adult AR of *A. palmata* in Mexico (Fig 3B) which are both of large size and reproductive [13]. The small fragment-based colonies surveyed in the present study (< 10 cm) are likely not reproductive and showed intermediate susceptibility.

The sexual recombination of traits in populations of larval recruits has been hypothesized to confer greater adaptability to restored populations [6, 30], compared to restoration interventions based on fragmentation of fixed genotype(s). Although the resistance of AR to bleaching found in this study could be considered consistent with this hypothesis, in four paired comparisons where we surveyed a given cohort of AR and colonies at the parental site from which that cohort of AR were bred (*P. strigosa* in St. Croix, *D. labyrinthiformis* in St. Croix and

Mexico; and two *A. palmata* corhorts in Mexico), all four parental populations showed high bleaching susceptibility (Fig 3B and 3C). This high susceptibility among parental populations is generally not consistent with the possibility that genetic inheritance and/or recombination from resistant parental genotypes accounts for offspring resistance.

The presence of resistant symbiont types is another plausible mechanism for observed resistance of AR. Recruits of all the species reported here acquire symbionts via horizontal transmission, i.e., from their environment rather than by maternal inheritance. This is understood to be a flexible process [31] with multiple potential sources for transmission. For example, *P. strigosa* recruits exposed to natal reef sediment and coral host fragments in laboratory tanks contained more diverse Symbiodinaceae communities than adult conspecifics at their natal site [32]. Additionally, outplanted AR of *O. faveolata* in the Florida Keys hosted complements of symbionts that were, at least initially, distinct from both their parental population and from co-occurring conspecific adults [33]. AR of *Acropora palmata* reared in land-based propagation efforts are capable of maintaining symbiosis predominantly with heat-tolerant *Durisdinium* [34] over multiple years after outplanting which could constitute an added intervention, though performance during bleaching events has yet to be tested. This enhanced diversity implies flexibility in the symbiont community composition of young corals that may confer the capacity for rapid adjustment to heat-tolerant assemblages in the case of environmental extremes.

## Conclusion

As previously hypothesized, sexually-propagated juvenile corals of six important reef-building species of less than five years of age were highly resistant to bleaching and related mortality during and within a month following the extreme marine heatwave in July-November 2023 (DHWs up to 25°C-weeks). Although bleaching resistance may wane with age, seeding can make an important contribution to coral population persistence assuming some assisted recruits reach reproductive maturity. As coral reefs throughout the globe are hit by increasingly frequent and intense marine heatwaves, restoration activities that include sexual reproduction and seeding can make an important contribution to sustain coral populations.

## Supporting information

**S1 Fig. Associations between colony size and Bleaching and Mortality Index (BMI) for each species used in restoration, with the negative trend for wild *A. palmata* being the only statistically significant trend.** Note that colony size was not consistently recorded across the data set.
(TIF)

**S1 Data.**
(CSV)

**S1 Table. Model results testing for species-specific differences in bleaching responses between assisted recruits and other comparison colonies (model form: Response ~ AR-Comparison \* Species), such that a negative estimate is a worse health status.** The cumulative link model (CLM) uses ordinal health status scores as the response variable, the GLM and GLMM use the Bleaching and Mortality Index as the response variable, and the GLMM in addition accounts for variations in response due to region, site, or population type (proxy of census method).
(DOCX)

## Acknowledgments

The dedicated staff and partners all of the authors' institutions that were involved in propagation and monitoring are gratefully acknowledged for enabling this study. Specifically, R. Tecalco Renería, E. Avila Pech, T. Doblado Speck, S. Tuijten, N. Le Trocquer, M. Davies, D. Gonzalez, R. Maronde, S. Orndorff, M. Alperstein, S. Bideau contributed to data collection. M. V. Grosso-Becerra is also gratefully acknowledged for programmatic contributions.

## Author Contributions

**Conceptualization:** Margaret W. Miller, Sandra Mendoza Quiroz, Anastazia T. Banaszak, Dirk Petersen.

**Data curation:** Margaret W. Miller, Sandra Mendoza Quiroz, Liam Lachs, Maria F. Villalpando.

**Formal analysis:** Margaret W. Miller, Liam Lachs.

**Funding acquisition:** Dirk Petersen.

**Investigation:** Sandra Mendoza Quiroz, Valérie F. Chamberland, Alexandra N. Gutting, Kelly R. W. Latijnhouwers, Francesca Virdis, Maria F. Villalpando.

**Methodology:** James R. Guest.

**Project administration:** Margaret W. Miller, Anastazia T. Banaszak.

**Resources:** Anastazia T. Banaszak, James R. Guest, Rita I. Sellares-Blasco, Francesca Virdis.

**Supervision:** Rita I. Sellares-Blasco.

**Writing – original draft:** Margaret W. Miller.

**Writing – review & editing:** Sandra Mendoza Quiroz, Liam Lachs, Anastazia T. Banaszak, Valérie F. Chamberland, James R. Guest, Alexandra N. Gutting, Kelly R. W. Latijnhouwers, Rita I. Sellares-Blasco, Francesca Virdis, Maria F. Villalpando, Dirk Petersen.

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
