## [Decision Letter · Decision Letter 0]

24 Jun 2024

PONE-D-24-14553Assisted sexual coral recruits show high thermal tolerance to the 2023 Caribbean mass bleaching eventPLOS ONE

Dear Dr. Miller,

Thank you for submitting your manuscript to PLOS ONE. After careful consideration, we feel that it has merit but does not fully meet PLOS ONE’s publication criteria as it currently stands. Therefore, we invite you to submit a revised version of the manuscript that addresses the points raised during the review process. Both reviewers have made good suggestions to improve the manuscript. In particular, please note the comments of Reviewer #2 regarding the need for more explicit caveats regarding the comparisons between restored and natural specimens. Your data sets should meet the assumptions of the statistical tests. If not, please revise those analyses. Please make sure that all of your data are fully available. See #3 below. There appear to be gaps of data (missing rows) in your Excel sheets. Please check those.

We look forward to receiving your revised manuscript.

Kind regards,

Erik V. Thuesen, Ph.D.

Academic Editor

PLOS ONE

Journal Requirements:

Reviewers' comments:

Reviewer's Responses to Questions

**Comments to the Author**

1. Is the manuscript technically sound, and do the data support the conclusions?

Reviewer #1: Yes

Reviewer #2: No

2. Has the statistical analysis been performed appropriately and rigorously? 

Reviewer #1: Yes

Reviewer #2: No

3. Have the authors made all data underlying the findings in their manuscript fully available?

Reviewer #1: Yes

Reviewer #2: No

4. Is the manuscript presented in an intelligible fashion and written in standard English?

Reviewer #1: Yes

Reviewer #2: Yes

5. Review Comments to the Author

Reviewer #1: Miller et al. have submitted a very compelling study indicating that larval recruits may be more bleaching tolerant than larger, conspecifics (or comparative populations). Although there are challenges within the data set, the authors do a substantial job working with the data that are available. This could, of course, lead to a more targeted study that controls for species, colony size, and environmental conditions to see how ubiquitous these results may be and when the ontogenetic shift may result in the loss of this trait.

The statistical approach is strong and comprehensively tackles it in several ways ensuring that the caveats are accounted for in the best way possible. The team also represents a breadth of expertise and a diverse geography.

I have listed some minor comments and suggestions below. There are no major issues to address.

Abstract

Line 43: why not ASR to capture all the words as in Koch et al.?

Line 50: change ‘in which’ to ‘when’ to better convey this occurred as a replacement when conspecifics were rare

Line 53: ‘hypothesized mechanisms were explored’ – the follow up sentence doesn’t convey a mechanism, just explores another type of coral and bleaching susceptibility that could be related to size or production source…think reword this to reflect different types of corals explored or something like that

Introduction

Line 69: I think its worth mentioning that there are additional steps AND it’s harder to accomplish AND there was a lot of R&D needed to do it even with the species we can use today.

Line 74/75: ‘long been’ is used twice in the same sentence…recommend altering one.

Line 77: change to ‘…juveniles have shown vulnerability…’

Lines 81-84: This could also be a result of differences in mass transfer and not related to juvenile vs adult life stage, which I think would be worth mentioning Effects of coral colony morphologies on mass transfer and susceptibility to thermal stress | Coral Reefs (springer.com)

Methods

Line 115: change to ‘but within four weeks of estimated peak heat stress’

Lines 134/135: change to ‘most appropriate comparison as both species and environmental exposure was matched at the same site’

Line 149: I am unsure of what ‘to remove the possible influence of assemblage structure on emergent trends’ means here

Line 149: there is a ‘Firstly’ here, but then on line 157 there is another ‘First’ and a ‘Secondly’ on line 162…seems like we need a different beginning to the sentence

Results and Discussion

Line 181: change ‘are’ to ‘were’

Line 182: change ‘retain’ to ‘retained’

Line 183: change ‘is’ to ‘was’

Line 203: something wrong with this sentence ending. Either words should be deleted or there are words missing.

Line 208: SCTLD, when spelled out, is not capitalized

Line 231: remove the word ‘significantly’ as there were no statistical tests comparing these

Line 246-253: I think this is the coolest conclusion, where you can control for size and still compare juveniles vs asexual frags. But, figure 3 seem more compelling and the text is sort of confusing as you are comparing CNAT vs Orbicella vs Apal. The figures though suggest you have within species comparisons.

Line 288/289: change to ‘AR survive preferentially in grooves…’

Line 312/313: change to ‘in the lab host more diverse…’

Section around Line 315/316, could add another interesting study, Long-term maintenance of a heterologous symbiont association in Acropora palmata on natural reefs | The ISME Journal | Oxford Academic (oup.com)

Reviewer #2: The manuscript by Miller et al describes the impacts of the 2023 mass coral bleaching event on corals generated through assisted reproduction deployed at 5 locations throughout the Caribbean region over the last ~12 years. They compare these impacts to reference populations at each site based on availability of conspecifics or co-occurring coral populations. While I think it is very important to document the impacts of the bleaching on natural and restored coral populations, and I really appreciate the effort the authors have made to find suitable comparison groups, my main concern is that I don’t think the comparisons are valid in some cases, given (1) species-level differences in heat tolerance, and (2) ontogenetic/size differences in heat tolerance. Given that it was nearly impossible to find similarly-sized natural conspecifics in the same locations as the outplants, the finding that ‘assisted sexual recruits had significantly lower prevalence of bleaching than conspecific corals’ (line 51/52) isn’t fair to make without clear and upfront caveats that the size-frequency distributions of AR and conspecific corals didn’t overlap (i.e. Fig S1).

I still believe that these data are important to publish and the lack of natural juvenile corals in these populations shouldn’t preclude it. I also appreciate that the authors were transparent that the summary analyses make ‘unbalanced comparisons among species, sites and colony origin’ (line 179), but it then begs the question – what’s the value of trying to make direct statistical comparisons? I think the text needs to be even more explicit about these caveats and the statistical comparisons should be limited in favour of simply reporting the observational data and then discussing the results and proposing mechanisms, potential implications and meaning, contextualized in the caveats.

My other concern is that these results are couched in the argument that AR populations should have greater resilience due to higher genotypic diversity via genetic recombination (line 40-41) and that their performance is compared to the ‘predicted resistance of sexually propagated juveniles’ (line 90). However, no genetic data for these outplanted populations are presented and there is very little evidence to date that demonstrates that restored coral populations are more genetically diverse than natural ones (although perhaps if no conspecifics exist at the site then by default it’s true). But, it is quite possible that inbreeding depression could also occur, reducing fitness. While there are great references and resources that can be used to consider ways to maximize adaptive potential (i.e. the Baums et al reference), I don’t think it’s supported here by genetic data. Were there specific ‘predictions’ (line 90) made beyond that the AR corals would be ‘more’ heat tolerant? Without a directly comparable reference population, it is difficult to evaluate whether this ‘resistance’ is in line with that ‘predicted’ so I would use caution in interpreting these results in the context of heat resistance/resilience.

Finally, I felt that the manuscript introduction was short and at times the introduction and discussion lacked continuity. I have pointed to some specific examples below that warrant attention.

Minor comments:

- Line 40: I suggest deleting the first sentence of the abstract. Firstly, it felt disjointed with the rest of the abstract and didn’t flow very well. Secondly, AR ‘could’ but not necessarily ‘would’ lead to more resilient restored populations as inbreeding depression could actually reduce resilience.

- Line 56 (And others): I found the use of the term ‘dosage’ in this context to be odd and maybe not needed.

- Line 59 (And 325): I suggest defining ‘demographic buffer’ here and in the conclusion. I’m not sure why maintaining high densities of juveniles (that are not yet reproductive) that become susceptible to bleaching as they grow would be beneficial, and reduce variance in vital rates (a la Hilde et al. 2020 TREE).

- Line 65: suggest ‘could’ rather that ‘should buy time’

- Line 74-75: suggest editing so ‘long been’ isn’t used twice in once sentence

- Line 75: I think a bridging sentence may be useful here to link the 1st and 2nd paragraphs. I also think the 2nd paragraph could benefit from additional examples and/or a discussion of potential mechanisms for why juveniles may be more (or less) susceptible than adults to various stressors. I understand it is unpacked in the discussion further, but I think the implications of being more or less susceptible to heat should be introduced here. The end of the 2nd paragraph could also benefit from a concluding statement.

- Line 91: I think this aim should make it clear what the susceptibility of AR is being compared to.

- Figure 1 caption: suggest changing ‘illustration’ to ‘image’ in (B).

- Table 2: fix shading formatting for CUR/CARMABI line

- Line 136: I recognize the need to identify some threshold to delineate adult and juvenile, but is there a justification for why >4 cm diameter was considered ‘adult’? Typically 5 cm is used as a cut-off, but even then it’s likely that colonies of many taxa aren’t yet reproductively mature at that size. Do you have a reference and/or justification for this?

- Line 147: I was confused by the statement that ‘a subset of data including only the six species used for restoration’ were used in the analyses because, according to table 1, there were wild assemblages at the sites used in some comparisons (table 1). So, were the wild populations limited to only those 6 taxa?

- Line 151: I’m not too familiar with the cumulative link model but a quick search indicated that it was a form of ordinal regression? Perhaps a brief justification/explanation of this approach would be useful since I don’t think it is that common of an analysis. Also, please report which program and packages were used for the analysis.

- Line 183: suggest changing ‘difference is’ to ‘difference was’ and checking for past tense throughout results and discussion

- Line 197-201: I am confused by this sentence. Is it suggesting that while within-species comparisons are most appropriate, most taxa around AR outplants were weedy so within-species comparisons were not possible? Also the sentence appears to be unfinished. Please edit.

- Paragraph of line 204: I think this paragraph requires more discussion. Does the point regarding SCTLD suggest that whole-colony mortality from disease may have been confused with bleaching-related mortality, and that this could disproportionately affect smaller colonies? Please expand and clarify this point.

- Line 217: suggest finishing this sentence by adding ‘…than adult colonies.’

- Line 266: suggest changing ‘this set of colonies were’ to ‘this set of colonies was’

6. PLOS authors have the option to publish the peer review history of their article (what does this mean?). If published, this will include your full peer review and any attached files.

Reviewer #1: **Yes: **Erinn Muller

Reviewer #2: No

---

## [Author Response · Author response to Decision Letter 0]

2 Aug 2024

Reviewer #1: Miller et al. have submitted a very compelling study indicating that larval recruits may be more bleaching tolerant than larger, conspecifics (or comparative populations). Although there are challenges within the data set, the authors do a substantial job working with the data that are available. This could, of course, lead to a more targeted study that controls for species, colony size, and environmental conditions to see how ubiquitous these results may be and when the ontogenetic shift may result in the loss of this trait.

The statistical approach is strong and comprehensively tackles it in several ways ensuring that the caveats are accounted for in the best way possible. The team also represents a breadth of expertise and a diverse geography.

I have listed some minor comments and suggestions below. There are no major issues to address.

Abstract

Line 43: why not ASR to capture all the words as in Koch et al.?

Koch et al. 2022 define the ASR acronym for Assisted Sexual Reproduction (the process, not the recruits themselves), which is somewhat different. Hence, we will retain the term AR (to refer to offspring, rather than the process of reproduction) for this paper.

Line 50: change ‘in which’ to ‘when’ to better convey this occurred as a replacement when conspecifics were rare

‘In which’ has been retained but this sentence has been revised. The full coral assemblages around the outplants were sampled as comparison populations to incorporate the same ecological exposure conditions, not just as replacements ‘when’ conspecifics were rare. 

Line 53: ‘hypothesized mechanisms were explored’ – the follow up sentence doesn’t convey a mechanism, just explores another type of coral and bleaching susceptibility that could be related to size or production source…think reword this to reflect different types of corals explored or something like that

This sentence has been deleted

Introduction

Line 69: I think its worth mentioning that there are additional steps AND it’s harder to accomplish AND there was a lot of R&D needed to do it even with the species we can use today.

This sentence has been revised/elaborated

Line 74/75: ‘long been’ is used twice in the same sentence…recommend altering one.

Done

Line 77: change to ‘…juveniles have shown vulnerability…’

This sentence has been deleted

Lines 81-84: This could also be a result of differences in mass transfer and not related to juvenile vs adult life stage, which I think would be worth mentioning Effects of coral colony morphologies on mass transfer and susceptibility to thermal stress | Coral Reefs (springer.com)

This possibility (and reference) is already mentioned in the discussion (now ln 326-327)

Methods

Line 115: change to ‘but within four weeks of estimated peak heat stress’

Done

Lines 134/135: change to ‘most appropriate comparison as both species and environmental exposure was matched at the same site’

Done

Line 149: I am unsure of what ‘to remove the possible influence of assemblage structure on emergent trends’ means here

Sentence has been revised

Line 149: there is a ‘Firstly’ here, but then on line 157 there is another ‘First’ and a ‘Secondly’ on line 162…seems like we need a different beginning to the sentence

Corrected 

Results and Discussion

Line 181: change ‘are’ to ‘were’

Line 182: change ‘retain’ to ‘retained’

Line 183: change ‘is’ to ‘was’

These were corrected

Line 203: something wrong with this sentence ending. Either words should be deleted or there are words missing.

Corrected

Line 208: SCTLD, when spelled out, is not capitalized

Corrected

Line 231: remove the word ‘significantly’ as there were no statistical tests comparing these

Done

Line 246-253: I think this is the coolest conclusion, where you can control for size and still compare juveniles vs asexual frags. But, figure 3 seem more compelling and the text is sort of confusing as you are comparing CNAT vs Orbicella vs Apal. The figures though suggest you have within species comparisons.

In the Bonaire in situ nursery, we did not have conspecific comparisons; AR were C.natans and fragments were Orbicella spp. We have new emphasized this in the text. Meanwhile, in the Dominican Republic there was indeed a conspecific comparison, which is why we chose to only depict these data in Fig 3. 

Line 288/289: change to ‘AR survive preferentially in grooves…’

Done

Line 312/313: change to ‘in the lab host more diverse…’

Done

Section around Line 315/316, could add another interesting study, Long-term maintenance of a heterologous symbiont association in Acropora palmata on natural reefs | The ISME Journal | Oxford Academic (oup.com)

This reference has now been included

Reviewer #2: The manuscript by Miller et al describes the impacts of the 2023 mass coral bleaching event on corals generated through assisted reproduction deployed at 5 locations throughout the Caribbean region over the last ~12 years. They compare these impacts to reference populations at each site based on availability of conspecifics or co-occurring coral populations. While I think it is very important to document the impacts of the bleaching on natural and restored coral populations, and I really appreciate the effort the authors have made to find suitable comparison groups, my main concern is that I don’t think the comparisons are valid in some cases, given (1) species-level differences in heat tolerance, and (2) ontogenetic/size differences in heat tolerance. Given that it was nearly impossible to find similarly-sized natural conspecifics in the same locations as the outplants, the finding that ‘assisted sexual recruits had significantly lower prevalence of bleaching than conspecific corals’ (line 51/52) isn’t fair to make without clear and upfront caveats that the size-frequency distributions of AR and conspecific corals didn’t overlap (i.e. Fig S1).

All statistical comparisons in the paper are restricted to con-specific comparisons (hence controlling for species-specific variation in heat tolerance). The cited sentence in the abstract has been modified to read ‘Assisted sexual recruits had significantly lower prevalence of bleaching impacts (overall pooled ~ 10%) than conspecific coral populations typically comprised of larger colonies (~ 60-100% depending on species).’

I still believe that these data are important to publish and the lack of natural juvenile corals in these populations shouldn’t preclude it. I also appreciate that the authors were transparent that the summary analyses make ‘unbalanced comparisons among species, sites and colony origin’ (line 179), but it then begs the question – what’s the value of trying to make direct statistical comparisons? I think the text needs to be even more explicit about these caveats and the statistical comparisons should be limited in favour of simply reporting the observational data and then discussing the results and proposing mechanisms, potential implications and meaning, contextualized in the caveats.

We only include one statistical analysis (and another supporting test without random effects in supplement), and it accounts for these biases. That addition of random effects turns a significant AR-Wild comparison for O. faveolata (see ordinal CLM) into a non-significant comparison (as all the difference is explained by Site (Bonaire/Mexico).

In any statistical analysis, one cannot include all possible covariates of the response variable. While there are very strong arguments against pooling all species together, there are more nuanced relationships between colony size and location versus bleaching. In this analysis we did not compare AR of one species to wild colonies of a different species, always restricting analyses within a single species. Notably, in our statistical comparison (Fig. 2B) the generalised linear mixed effect model does account for variability in bleaching due to possible confounding effects of region, site, and comparison population type (a proxy for census method). Importantly, if we do not account for any of these possible confounding factors, we even see a significant difference in tolerance for Orbicella faveolata, but once these confounding effects are included in the model, it increases the uncertainty (confidence intervals) in the AR-Wild bleaching comparison (Fig. 2B) and that group is no longer significantly different. For O. faveolata, that is because the two comparison groups come from two different countries (assisted corals from Bonaire vs wild corals from Mexico).

My other concern is that these results are couched in the argument that AR populations should have greater resilience due to higher genotypic diversity via genetic recombination (line 40-41) and that their performance is compared to the ‘predicted resistance of sexually propagated juveniles’ (line 90). However, no genetic data for these outplanted populations are presented and there is very little evidence to date that demonstrates that restored coral populations are more genetically diverse than natural ones (although perhaps if no conspecifics exist at the site then by default it’s true). But, it is quite possible that inbreeding depression could also occur, reducing fitness. While there are great references and resources that can be used to consider ways to maximize adaptive potential (i.e. the Baums et al reference), I don’t think it’s supported here by genetic data. Were there specific ‘predictions’ (line 90) made beyond that the AR corals would be ‘more’ heat tolerant? Without a directly comparable reference population, it is difficult to evaluate whether this ‘resistance’ is in line with that ‘predicted’ so I would use caution in interpreting these results in the context of heat resistance/resilience.

Acknowledged that we are not presenting any genetic data (our statements were simply based on general principles of sexual recombination as articulated in Baums et al. 2019). We have moderated the wording in both of these sections, avoiding ‘prediction’ terminology. 

Inbreeding depression certainly could occur, but is generally an unlikely outcome in single-generation breeding among large populations of unrelated individual (which have historically been the case for the species in question) as long as general precautions are taken (e.g., incorporating as many parents as possible in larval cohorts and not outplanting offspring adjacent to parents; Baums et al. 2019). Inbreeding depression becomes a much greater concern in breeding among ‘small’ populations. We are likely reaching a point where certain Caribbean coral populations are reaching ‘small’ status (e.g., A.palmata or D.cylindrus in Florida) and require much more careful breeding within genetic management plans to avoid – and such planning is currently underway for the Florida population of A.palmata (Rodriguez-Clark et al. 2023) but is beyond the scope for this paper.

Finally, I felt that the manuscript introduction was short and at times the introduction and discussion lacked continuity. I have pointed to some specific examples below that warrant attention.

Minor comments:

- Line 40: I suggest deleting the first sentence of the abstract. Firstly, it felt disjointed with the rest of the abstract and didn’t flow very well. Secondly, AR ‘could’ but not necessarily ‘would’ lead to more resilient restored populations as inbreeding depression could actually reduce resilience.

Sentence modified 

- Line 56 (And others): I found the use of the term ‘dosage’ in this context to be odd and maybe not needed.

Changed to ‘exposure’ throughout

- Line 59 (And 325): I suggest defining ‘demographic buffer’ here and in the conclusion. I’m not sure why maintaining high densities of juveniles (that are not yet reproductive) that become susceptible to bleaching as they grow would be beneficial, and reduce variance in vital rates (a la Hilde et al. 2020 TREE).

The term ‘demographic buffer’ has been removed from the manuscript and the intended meaning re-phrased.

- Line 65: suggest ‘could’ rather that ‘should buy time’

Revised wording here

- Line 74-75: suggest editing so ‘long been’ isn’t used twice in once sentence

Done

- Line 75: I think a bridging sentence may be useful here to link the 1st and 2nd paragraphs. I also think the 2nd paragraph could benefit from additional examples and/or a discussion of potential mechanisms for why juveniles may be more (or less) susceptible than adults to various stressors. I understand it is unpacked in the discussion further, but I think the implications of being more or less susceptible to heat should be introduced here. The end of the 2nd paragraph could also benefit from a concluding statement.

A sentence has been included here articulating hypotheses for bleaching resistnace

- Line 91: I think this aim should make it clear what the susceptibility of AR is being compared to.

This sentence now include reference to the Comparison population types.

- Figure 1 caption: suggest changing ‘illustration’ to ‘image’ in (B).

Done

- Table 2: fix shading formatting for CUR/CARMABI line

We are not sure about the meaning here. Shading in the table was intended to help the reader distinguish the rows related to different regions. Hence, the rows for St. Croix and Bonaire are shaded, those for MEX, CUR/CARMABI, and the DR are not.

- Line 136: I recognize the need to identify some threshold to delineate adult and juvenile, but is there a justification for why >4 cm diameter was considered ‘adult’? Typically 5 cm is used as a cut-off, but even then it’s likely that colonies of many taxa aren’t yet reproductively mature at that size. Do you have a reference and/or justification for this?

The 4 cm cutoff was chosen to be consistent with widespread monitoring protocols that are widely applied in the Caribbean (e.g. AGRRA and NOAAs national monitoring program). A sentence has been added to articulate this (Ln 145-148).

- Line 147: I was confused by the statement that ‘a subset of data including only the six species used for restoration’ were used in the analyses because, according to table 1, there were wild assemblages at the sites used in some comparisons (table 1). So, were the wild populations limited to only those 6 taxa?

This paragraph has been substantially revised to clarify that we collected data on all species in wild assemblages, and qualitatively compared the entire sample, while statistical modeling and comparisons were limited to within-species.

- Line 151: I’m not too familiar with the cumulative link model but a quick search indicated that it was a form of ordinal regression? Perhaps a brief justification/explanation of this approach would be useful since I don’t think it is that common of an analysis. Also, please report which program and packages were used for the analysis.

This is a common Ordinal regression method. The R-packages are now specified for each analysis

- Line 183: suggest changing ‘difference is’ to ‘difference was’ and checking for past tense throughout results and discussion

Done

- Line 197-201: I am confused by this sentence. Is it suggesting that while within-species comparisons are most appropriate, most taxa around AR outplants were weedy so within-species comparisons were not possible? Also the sentence appears to be unfinished. Please edit.

Incomplete sentence has been revised. This paragraph emphasizes the important contribution of AR in extant Caribbean coral communities, given the rarity/absence of natural recruitment by reef-building species.

- Paragraph of line 204: I think this paragraph requires more discussion. Does the point regarding SCTLD suggest that whole-colony mortality from disease may have been confused with bleaching-related mortality, and that this could disproportionately affect smaller colonies? Please expand and clarify this point.

Yes, the caveat was intended to acknowledge that full colony bleaching mortality estimates might have been affected by artifacts – specifically, small colonies may die more quickly (and hence be undetectable after colonization) than large colonies, thus underestimating bleaching impact on AR relative to larger colonies. Conversely, SCTLD mortality might have been lumped with bleaching mortality, hence inflating it, in our data s

---

## [Editor Report · Decision Letter 1]

14 Aug 2024

PONE-D-24-14553R1

Assisted sexual coral recruits show high thermal tolerance to the 2023 Caribbean mass bleaching event

PLOS ONE

Dear Dr. Miller,

Thank you for submitting your revised manuscript to PLOS ONE. After careful consideration, we feel that it has merit but does not fully meet PLOS ONE’s publication criteria as it currently stands. PLOS ONE does not copy-edit manuscripts. Therefore, we invite you to submit a revised version that addresses the two minor points below. 

We look forward to receiving your revised manuscript.

Kind regards,

Erik V. Thuesen, Ph.D.

Academic Editor

PLOS ONE

Journal Requirements:

**Additional Editor Comments:**

1) On line 121: correct to ‘NOAA’s’. 

2) Please clean up the literature cited. This includes 1) making sure that lower case and upper case are correct in article titles, 2) that all genus and species names are Italicized, and 3) that abbreviations of journal names are correct. 

---

## [Author Response · Author response to Decision Letter 1]

15 Aug 2024

Requested corrections have been made

---

## [Editor Report · Decision Letter 2]

19 Aug 2024

Assisted sexual coral recruits show high thermal tolerance to the 2023 Caribbean mass bleaching event

PONE-D-24-14553R2

Dear Dr. Miller,

We’re pleased to inform you that your manuscript has been judged scientifically suitable for publication and will be formally accepted for publication once it meets all outstanding technical requirements.

Kind regards,

Erik V. Thuesen, Ph.D.

Academic Editor

PLOS ONE
---

## [Editor Report · Acceptance letter]

23 Aug 2024

PONE-D-24-14553R2 

PLOS ONE

Dear Dr. Miller, 

I'm pleased to inform you that your manuscript has been deemed suitable for publication in PLOS ONE. Congratulations! Your manuscript is now being handed over to our production team.

Kind regards, 

on behalf of

Dr. Erik V. Thuesen 

Academic Editor

PLOS ONE